# Nuclear Magnetic Resonance Therapy Modulates the miRNA Profile in Human Primary OA Chondrocytes and Antagonizes Inflammation in Tc28/2a Cells

**DOI:** 10.3390/ijms22115959

**Published:** 2021-05-31

**Authors:** Bibiane Steinecker-Frohnwieser, Birgit Lohberger, Nicole Eck, Anda Mann, Cornelia Kratschmann, Andreas Leithner, Werner Kullich, Lukas Weigl

**Affiliations:** 1Ludwig Boltzmann Institute for Arthritis and Rehabilitation, 5760 Saalfelden, Austria; bibiane.frohnwieser@medunigraz.at (B.S.-F.); nicole.eck@medunigraz.at (N.E.); lbirehab@aon.at (W.K.); 2Department of Orthopedics and Trauma, Medical University Graz, 8036 Graz, Austria; andreas.leithner@medunigraz.at; 3Division of Special Anaesthesia and Pain Medicine, Medical University of Vienna, 1090 Vienna, Austria; anda.mann@meduniwien.ac.at (A.M.); cornelia.kratschmann@meduniwien.ac.at (C.K.); lukas.weigl@meduniwien.ac.at (L.W.)

**Keywords:** NMRT, osteoarthritis, miR, MMPs, inflammation

## Abstract

Nuclear magnetic resonance therapy (NMRT) is discussed as a participant in repair processes regarding cartilage and as an influence in pain signaling. To substantiate the application of NMRT, the underlying mechanisms at the cellular level were studied. In this study microRNA (miR) was extracted from human primary healthy and osteoarthritis (OA) chondrocytes after NMR treatment and was sequenced by the Ion PI Hi-Q™ Sequencing 200 system. In addition, T/C-28a2 chondrocytes grown under hypoxic conditions were studied for IL-1β induced changes in expression on RNA and protein level. HDAC activity an NAD^(+)^/NADH was measured by luminescence detection. In OA chondrocytes miR-106a, miR-27a, miR-34b, miR-365a and miR-424 were downregulated. This downregulation was reversed by NMRT. miR-365a-5p is known to directly target HDAC and NF-ĸB, and a decrease in HDAC activity by NMRT was detected. NAD^+^/NADH was reduced by NMR treatment in OA chondrocytes. Under hypoxic conditions NMRT changed the expression profile of HIF1, HIF2, IGF2, MMP3, MMP13, and RUNX1. We conclude that NMRT changes the miR profile and modulates the HDAC and the NAD^(+)^/NADH signaling in human chondrocytes. These findings underline once more that NMRT counteracts IL-1β induced changes by reducing catabolic effects, thereby decreasing inflammatory mechanisms under OA by changing NF-ĸB signaling.

## 1. Introduction

Osteoarthritis (OA) represents the most common form of degenerative joint disease and is characterized by alterations of the cartilage extracellular matrix composition. The underlying degradation of the articular cartilage associated with the limited repair capacity leads to joint integrity disruption and progressive irreversible dysfunction [1]. Although, therapeutic agent concepts coupled with surgical techniques have been developed to treat patients with OA, there are no real effective medical therapies to prevent cartilage destruction and the associated changes in joints and bone. Interestingly, there is an attempt to use therapeutic nuclear magnetic resonance therapy (NMRT) in OA treatment. In clinical observations NMRT has been demonstrated to cause positive effects when applied to treat painful disorders of the musculoskeletal system [2]. The main outcomes were the pain relief of patients with low back pain and increased function in finger joint mobility in patients with hand OA [3,4]. While less was known about NMRT and its function at the cellular level [5], our previous work depicts positive effects of NMRT on the subthreshold inflammatory mechanisms of cultured chondrocytes and counteracting the interleukin-1β (IL-1β)-induced reduction in ATP and nuclear factor ‘kappa-light-chain-enhancer’ of activated B-cells (NF-ĸB) activity [6]. Focusing on NMRT influencing modulators of OA chondrocytes physiology like microRNAs (miR), regulation at the transcriptional level or hypoxic conditions might be beneficial.

MiRs represent a family of endogenous small non-coding RNAs and can lead under abnormal expression to impairment of normal cellular function [7]. Currently, accumulating studies show a vast diversity of miRs to act as regulators in the pathogenesis of OA. Differential expression of pro-inflammatory mediators in OA, like IL-1β or tumor necrosis factor-α (TNF-α) are regulated by miRs. IL-1β induced reprogramming of chondrocytes leading to the production of catabolic enzymes such as matrix metalloproteinases (MMP)1, MMP3, MMP13, cyclooxygenase 2 (COX2) or upregulation of transcription factors like runt-related transcription factor 2 (RUNX2) or hypoxia-inducible factor-2α (HIF2α) has been shown to be regulated by miR targeting inflammatory pathways [8]. Furthermore, as another keyplayer in OA, NF-kB provokes the abnormal production of proinflammatory cytokines (IL-1β/TNFα) and pro-catabolic mediators, including aggrecanases and MMPs inducing cartilage degradation [9]. The dysfunctional behavior of cells might be corrected by the right treatment methods, shutting down the inflammatory pathways and the production of matrix destroying enzymes.

The deregulation of catabolic and anabolic processes in OA can also be found at the gene transcription level. Histone modification plays a pivotal role in controlling gene expression, being involved in a wide spectrum of diseases including OA [10]. The group II HDAC4 has altered expression and activities in OA while it is significantly downregulated by HDAC inhibitors and specific siRNAs [11]. Group III HDACs, the sirtuins, depend on NAD(+) and link their enzymatic activity directly to the energy status of the cell via the cellular NAD(+):NADH ratio. Interestingly, HIF-2α has been shown to activate the NAMPT-NAD(+)-SIRT axis in chondrocytes and thereby contribute to the pathogenesis of OA [12].

Given that normal articular cartilage is hypoxic, chondrocytes have a specific and adapted response to low oxygen environment. Hypoxia promotes redifferentiation and counteracts dedifferentiation and is considered to have a positive influence on the healthy chondrocyte phenotype and cartilage matrix formation. However, HIFs have been implicated in the pathogenesis of OA [13]. Our study aims to target three regulatory mechanisms that have dysfunctional OA affect and that to a modest extent can be regulated by NMRT.

## 2. Results

### 2.1. Differences in miR Expression in OA Cells and the Influence of NMRT

The miR sequencing of HC and OA chondrocytes, both non-treated and treated with NMRT for 5 h, yielded 441 miRs after normalization to be differentially expressed. The principal component analysis demonstrated a clear separation of HC and OA collectives independent from the treatment paradigm (Figure 1A). The differential expression data were also used to outline single miRs up- or down regulated by OA in a volcano plot (Figure 1B). The comparison of untreated HC and untreated OA chondrocytes yielded 53 miRs with significantly changed expression. The extent of OA induced deregulation of miRs expression is pictured by a heat map (Appendix A). We further investigated whether the treatment of OA chondrocytes with NMRT can counteract this OA induced miR variation. The log2 fold change analysis of the NMRT effect on OA cells when compared to HC cells resulted in slight differences represented by the heat map (Appendix A). In addition, from these data 19 miRs showed at least 30% difference between the fold change of HC versus OA chondrocytes and HC versus OA chondrocytes plus NMRT (Appendix A). t For the chondrocyte metabolism, miR-122-5p [14], miR-203a-3p [15], miR-24-1-5p [16], miR-365a-5p [17,18], miR-4284 [19], miR-502-5p [20] are relevant, whereas miR-34b-5p and miR-210-3p are involved in the NF-ĸB pathway [8,13]. When we evaluated the NMRT effect on the miR profile of HC and OA chondrocytes, respectively, we detected a small number of miRs to be statistical significantly changed (Figure 1C,D). Within this handful of miRs again miR-106a-5p (Figure 1E), miR-24-1-5p, miR-365a-5p, and miR-502-5p popped up to be influenced by NMRT. The expression of specific miRs were validated by qPCR and slight, moderate changes have been detected. Although no significant change between HC versus OA chondrocytes for miR-106a-5p was detected, a trend in expression was observed by treating HC or OA chondrocytes with NMRT (Figure 1F). The slight and moderate reduction in expression of miR-27a-3p, miR-34b-5p, miR-365a-5p, and miR-424-5p in OA chondrocytes compared to HC was faintly reversed by the treatment of OA chondrocytes with NMRT. By contrast, NMRT counteracted the increase in the expression of miR-210-3p (Figure 1G).

### 2.2. NMRT Affects Players Involved in miRNA Regulated Signaling

MiR-24-1-5p is involved in MMP1, MMP13, and p16INK4a regulation while miR-365a-3p regulates HDAC and NF-ĸB activity [17,18]. Therefore, we tested HC and OA chondrocytes for their HDAC4 activity plus COX2 and CDK4 expression under the influence of NMRT (Figure 2). HDAC4 expression was significantly reduced by NMRT in OA cells (Figure 2A). A comparison in expression of OA cells with control HC cells revealed a slight but significant upregulation (fold change av/std: 2.26 ± 0.65) of the HDAC4 expression by OA (HC/OA).This increase turned out to be reduced by 32 ± 24.9% when both HC and OA cells were treated with NMRT (Figure 2D; HC^NMRT^/OA^NMRT^). Interestingly, when both cell types were treated with NMRT, HDAC4 mRNA reduction could be strengthened. Similarly, at the activity level the HDAC4 activity increased in OA cells and was reduced to normal levels when OA chondrocytes were NMRT treated (Figure 2B). Trichostatin exposure of cells was used as control for the HDAC activity assay. When using NMRT treatment the concentration dependent block of the HDAC4 activity by trichostatin was significantly reduced in HC and OA cells (Figure 2C). Investigations concerning COX2 led to the finding of a COX2 overexpression in OA cells that could not be reversed by NMRT to the level of healthy HC cells (Figure 2D). The ratio of phosphorylated pCDK4/CDK4 was increased in OA cells but NMRT had no influence (Figure 2E).

### 2.3. The Cofactor System NAD^+^/NADH Is Influenced by NMRT

The NAD^(+)^/NADH ratio regulates the activity of NAD-dependent histone deacetylases (HDACs). We analyzed the influence of NMRT on changes of NAD^(+)^ and NADH, its ratio and the influence of IL-1β on HC compared to OA cells. A reduction in NAD^(+)^/NADH in both types of cells after the 5 h treatment with NMRT was observed (Figure 3A). Preconditioning of both types of cells with IL-1β increased NAD^(+)^/NADH in HC chondrocytes by 1.45 ± 0.27-fold (*p* ≤ 0.001), in OA cells by 3.17 ± 1.55-fold (*p* ≤ 0.001) and in OA chondrocytes treated with NMRT only by 1.88 ± 0.86-fold (*p* ≤ 0.01) (Figure 3B). The IL-1β effect was more pronounced in OA cells showing a statistically significant difference when compared to IL-1β regulated NAD^(+)^/NADH increase in HC cells (*p* ≤ 0.05). NMRT downregulated this observation by 41 ± 27% (*p* ≤ 0.05) and therefore nearly restoring values seen in HC cells.

A more detailed analysis by separate detection of NAD^(+)^ and NADH revealed differences in NADH and not NAD^(+)^ to be responsible for the co-factor increase in OA chondrocytes (Figure 3C). The highly significant elevated levels of NADH in OA cells when compared to HC cells could be reversed by NMRT. Furthermore, the effect by IL-1β that increases the NAD+/NADH ratio in OA cells was also reversed by NMRT treatment (Figure 3D).

### 2.4. Hypoxia Induced Changes in the Expression of T/C-28a2 Cells

T/C-28a2 cells were grown under hypoxic conditions by O_2_ displacement via nitrogen (N_2_). The O_2_ was replaced within 5 to 20 min (Appendix A) and the O_2_ concentration was constantly monitored with an oxygen sensor. Under normoxic conditions treatment of T/C-28a2 cells with IL1-β/TNFα resulted in a slight but non-significant reduction of healthy cells with a corresponding increase in the number of late apoptotic cells (Appendix A). Hypoxic conditions increased the fraction of early and late apoptotic cells. NMR treatment had no influence on the apoptotic behavior of the cells (Appendix A).

To find out if regulators involved in OA are influenced by hypoxia and if those might be targets for NMRT we tested the expression of several candidates by qPCR (Figure 4A). While IGF2, IGFBP3, HIF2, VEGF, and PDGF were increased in expression by hypoxic conditions, HIF1 was slightly decreased. Growth factors such as TGFβ, FGF, EGF, and RUNX2, IL-1β and TNFα were not changed. The fold changes in expression due to hypoxic conditions of the relevant targets were 7.22 ± 2.35 for IGF2; 29.75 ± 16.9 for IGFBP3; 0.67 ± 0.09 for HIF1; 3.48 ± 1.11 for HIF2; 0.11 ± 0.05 for MMP3; 1.47 ± 0.62 for MMP13; 6.74 ± 3.99 for VEGF, and finally 2.12 ± 0.58 for PDGF (Figure 4B).

### 2.5. NMRT Influences IL-1β/TNFα Induced Changes in T/C-28a2 Chondrocytes under Hypoxic Conditions

Under hypoxic conditions NMRT slightly increased the growth factors PDGF, VEGF, and IGFBP3. The two transcription factors HIF1 and RUNX2 and also IGF2 showed small but significant decreases by NMRT (Figure 5A). The influence of NMRT was tested on IL-1β/TNFα induced inflammatory T/C-28a2 cells. IL-1β/TNFα reduced the expression of HIF1 and increased HIF2, IGF2, MMP3, MMP13, and RUNX1. The induced effects in expression were reversed when cells were treated with NMRT (Figure 5B–G; Appendix A).

## 3. Discussion

OA is the most abundant joint disease and is associated with loss of function and pain. There is no cure up to date, and the clinical treatment of OA is currently unsatisfactory. To understand the pathogenesis of OA especially on the molecular level is imperative for developing new therapeutic strategies. Therefore, this study was designed to identify possible targets of NMRT which is considered as a therapeutic tool to treat OA. Currently, accumulating studies indicate miRs as crucial regulators of various biological processes and cellular functions. Aberrant miR expression is closely associated with cartilage degeneration, degradation, and regeneration in the pathogenesis of OA [8]. In our study, miR sequencing revealed 19 miRs to be influenced by OA and/or NMRT. Seven of these miRs (miR-27a-3p, miR-210-3p, miR-34b-5p, miR-365a-3p, miR-424-5p, miR-502-5p, and miR-106a-5p) responded with a mild change in expression to the treatment with NMRT.

Although, there are clinical studies where no substantial effect had been shown [21], from our previous studies we know that NMRT counteracts IL-1β induced effects involving NF-ĸB signaling in chondrocytes [6]. MiR-210, slightly changed, has been described as inhibitor of the NF-ĸB signaling pathway by targeting death receptor 6 [22]. Even though miR-34a is described to be involved in chondrocytes death causing OA progression through DLL1 and modulation of the PI3K/AKT pathway, less is known of miR-34b and OA chondrocyte development [23]. Interestingly, a direct targeting of IGFBP2 by miR-34b for mediating the proliferation and differentiation of myoblasts has been described [24]. Furthermore, IGFs are key regulators of matrix homeostasis in articular cartilage, and it has been proposed that dysregulation of metabolism during OA is due to IGF insensitivity [25]. IGFBPs can either promote or inhibit IGF activity or may act independently of IGFs. IGFBP3 has been shown to suppress NF-κB activation and chemokine secretion which were induced by TNFα in OA fibroblast-like synoviocytes (FLS), and also sensitized OA FLS to TNFα-induced apoptosis [26,27]. In our study chondrocytes treated with NMRT exhibited increases in IGFBP3, miR-34b expression and show a significant down-regulation of IL-1β induced IGF2, thereby one might assume linked regulations by NMRT.

Ji et al. (2018) demonstrated that, when injected intra-articularly in OA mice, miR-106a-5p ameliorates the progression of OA by inhibited the GLI-similar 3 (GLIS3) production [28]. NMRT reversed the downregulation of miR-106a-5p in OA compared to HC cells and directly increased the expression of miR-106a-5p in both cell types. This could imply that NMRT restores the suggested chondro-protective role of miR-106.

In articular chondrocytes it has been shown that inhibition of miR-365 down regulated the IL-1β induced expression of MMP13 and RUNX2 [29], but also that overexpression of miR-365 suppressed IL-1β induced up regulation of HIF2α and catabolic factors such as COX2, MMP1, MMP3, and MMP13. In our study NMRT slightly reduces the IL-1β induced up-regulation of MMP3, MMP13, RUNX2, and HIF2α, whether this effect is due to miR-365 modulation, remains unclear. Strong evidence for NMRT influencing MMPs is the observed reversal of the down regulation of miR-27 in OA cells, for it has been demonstrated that the increase of MMP13 correlated with down-regulation of miR-27 in OA chondrocytes [30]. Additionally, Wang et al., 2019 showed that miR-27a-3p mimics were able to abolish the effects of the long noncoding RNA FOXD2-AS1 overexpression on cell proliferation, inflammation, and ECM degradation in chondrocytes [31].

MiR-502 by NMRT once more substantiates these findings due to the discussion that miR-502-5p may exhibit a protective effect on IL-1β-induced chondrocyte damage by targeting TRAF2 and inhibition of the NF-κB signaling pathway. The upregulation of miR-502 by NMRT in OA cells thus corroborates our previous data that NMRT inhibits the NF-κB signaling [6]. In our study the expression of miR-140, characteristically expressed in cartilage, was increased but unchanged by NMRT. MiR-140 and miR-365 among others repress expression of HDAC4 leading to a significant suppression of IL-1β induced up-regulation of at least MMP1, MMP3, and MMP13 [32,33]. NMRT by influencing miR expression profile might therefore modulate MMP expression via the suppression in HDACI/II activity and HDAC4 expression. Decrease of HDAC activity in OA cells and the observed opposite effects under trichostatin (TSA) might contribute to a NMRT induced chondrocyte protection. In addition, we suggest that NMRT might modulate chondrocytes in advance so that TSA cannot play off its full potential. Furthermore, NMRT down regulating NAD^(+)^/NADH might point to NMRT and its intervention with the nicotinamide phosphoribosyltransferase (NAMPT)/NAD^(+)^/SIRT axis which is involved in OA cartilage destruction possibly via HIF2α down regulation [12].

From our former studies we know that NMRTs impact on chondrocytes is rather weak under normoxic conditions. To mimic more closely physiologic conditions hypoxic culture of human chondrocytes was established and resulted in increased matrix accumulation, and HIF expression. HIF is associated with the downregulation of hypertrophic markers and degradative enzymes as well as with redifferentiation of the chondrocytes [34]. Other than in our previous study under normoxic conditions [6], the expression of IGF2, IGFBP3 as well as MMP3, VEGF, and PDGF under hypoxic conditions was changed by NMRT underlining the importance of appropriate culture conditions for chondrocytes.

## 4. Materials and Methods

### 4.1. Cell Culture

Commercially available cells were used for this work. Therefore, no ethics vote was necessary. Human healthy chondrocytes (HC) and human osteoarthritis chondrocytes (OA) were purchased (Cell Application, Inc., San Diego, CA, USA). Cells were cultivated and grown in human chondrocyte media (HC Growth Medium, Cell Application, Inc.). The immortalized human chondrocyte cell line T/C-28a2 (kindly provided by Prof. M.B. Goldring, Harvard Institute of Medicine, Boston, MA, USA) is a common tool in cartilage research [35,36]. Cells were cultured in Dulbeco’s modified eagle’s medium (DMEM high glucose; GIBCO, Invitrogen, Darmstadt, Germany) supplemented with 10% fetal bovine serum (FBS), 1% L-glutamine, 100 units/mL penicillin, 100 μg/mL streptomycin (all GIBCO), and 0.25 μg amphotericin B (PAA Laboratories, Pasching, Austria). If required cells were supplemented with IL-1β (10 ng/mL, Sigma-Aldrich, Vienna, Austria) and TNFα (5 ng/mL, Thermo Fisher Scientific, Waltham, MA, USA) 24 h after seeding [37].

### 4.2. Nuclear Magnetic Resonance Therapy (NMRT)

NMRT treatment was applied by a specific device adapted for cell cultures and producing a magnetic field of 0.23 mT and an electromagnetic field of approximately 100 kHz (MedTec Company, Wetzlar, Germany). The NMRT treatment lasted for 5 h over a period of three days and was performed at room temperature outside the incubator. The untreated control cells were kept under comparable conditions. HEPES containing culture medium was used to keep the pH value within the neutral range.

### 4.3. Cell Culture under Hypoxic Conditions

At a density of 10.,000 cells per cm^2^ T/C-28a2 cells were cultured in 6 well fluorocarbon film imaging plates (Zell-Kontakt, Göttingen, Germany), put in an airtight plastic box and perfused with nitrogen to replace the oxygen. The film bottom of the plates enabled a high gas exchange rate and O_2_ was measured with an oxygen sensor at least every 6 to 12 h within the 3 days of NMRT treatment period. The nitrogen perfusion was repeated if the oxygen value went near 5%, although the concentration of O_2_ was mostly stable between 1% and 4.5%. The primary gas exchange took about 20 min, not only in the box but also within the well of the cell culture plate.

### 4.4. miR Sequencing

RNA was extracted from HC and OA cells with or without NMRT treatment using RNeasy Mini Kit (Qiagen, Hilden, Germany). After photometric quantification at 260/280 nm (NanoDrop, Thermo Fisher Scientific) up to 1 µg of RNA was used as input for the Ion Total RNA-Seq Kit v2 following the small RNA workflow (Thermo Fisher Scientific). Ion adapters were ligated to RNA molecules and reverse transcription was performed with SuperScript III enzyme mix. Size selected cDNA was amplified in 14 cycles of PCR. The purified products were analyzed by the Agilent Bioanalyzer to verify the molecule length (94 bp to 114 bp). Sequencing was performed on Ion Proton System for next generation sequencing using Ion PI Hi-Q Sequencing 200 Kit (Thermo Fisher Scientific) to a depth of approximately 7 million reads per sample. Signal processing and base calling was performed using Torrent Suite version 5.6. BAM files that were transferred to a Linux cluster computing environment. CAP-miRSeq software was used to derive miRNA counts from NGS by mapping to miRBase version 21. Counts were normalized and differential expression was calculated using edgeR [38,39]. From the derived data a principal component analysis to analyze the interrelationships among HC and OA groups independent of treatment conditions was performed. To quickly identify changes, a volcano plot from the same data set was calculated [40].

### 4.5. miRs Expression

Total RNA, including miR, was isolated from HC and OA control- and treated-cells (NMRT for 5 h over a period of 3 days) using an miRNeasy Mini Kit according to manufacturer’s protocol (Qiagen). The concentration of total RNA was measured, and the integrity of isolated RNA was assessed on an Agilent 2100 Bio Analyzer giving RIN ranges from 9–10. Subsequently, 200 ng of total RNA was reverse transcribed by using the miRCURY LNA Universal RT microRNA PCR starter Kit (Exiqon, Qiagen). UniSp6 spike-in control as well as the references miR103a-3p and miR-191were used. The cDNA testing for miR-424, miR-24, miR-34, und miR-502 was used at a 1:20. For detection of genes of interest, the MicroRNA LNA PCR primer sets were used (Exiqon). Real time PCR was on CFX96 touch (BioRad, Hercules, CA, USA). Relative quantification of expression was calculated using the ∆∆Ct-method.

### 4.6. Reverse Transcription Polymerase Chain Reaction (RT-PCR)

Total RNA was isolated from treated and untreated T/C-28a2 cells with the RNeasy Mini Kit and DNase-I treatment according to the manufacturer’s manual (Qiagen). One µg RNA was reverse transcribed via the iScript cDNA Synthesis Kit (BioRad). Sequences for PCR primers and amplicon size are given in Appendix A. QuantiTect primer assays (Qiagen) were used for HIF1 and HIF2. Primer sequences were derived from the Primerbank database (http://pga.mgh.harvard.edu/primerbank; accessed on 1 June 2016) or from PubMed Primerblast (Appendix A). Reactions were performed in duplicates. Amplification was achieved with SsoAdvance Universal SYBR Green Supermix (BioRad) on a Realplex Mastercycler (Eppendorf, Hamburg, Germany). Each qPCR run consisted of 40 cycle of a standard 3-step PCR temperature protocol (annealing temperature 60 °C) followed by melting curve protocol to confirm a single gene-specific peak. Relative quantification of expression was obtained by the ΔΔCt method based on the geometric mean of the internal controls glyceraldehyde 3-phosphate dehydrogenase (GAPDH), aldolase, and eukaryotic translation initiation factor 3 (ETIF). For IL-1β stimulation ΔCt values of the unstimulated cells functioned as control (*n* = 6).

### 4.7. Western Blot Analysis

Whole cell protein extracts were prepared with lysis buffer (50 mM Tris-HCl pH 7.4, 150 mM NaCl, 1 mM NaF, 1 mM EDTA, 1% NP-40, 1 mM Na3 VO4, protease inhibitor cocktail; Sigma Aldrich), subjected to SDS-PAGE and blotted onto Amersham Protran Premium 0.45 μM nitrocellulose membranes (GE Healthcare Life Science, Little Chalfont, UK). Protein concentrations were determined using the Pierce BCA Protein Assay Kit (Thermo Fisher Scientific). Primary antibodies were purchased from Cell Signaling Technology (Leiden, The Netherlands). Blots were developed using horseradish peroxidase-conjugated secondary antibody (Dako, Jena, Germany) at room temperature for 1 h and the Amersham ECL prime detection reagent (GE Healthcare), in accordance to manufacturers protocol. Chemiluminescence signals were detected with the ChemiDocTouch Imaging System (BioRad) and images were processed with the ImageLab 5.2 Software (BioRad). HDAC4 detection signals were normalized to the β-actin signal, for CDK4 the ratio between phosphorylated and non-phosphorylated protein was calculated (*n* = 4).

### 4.8. HDAC I/II Activity Measurement

In one experiment HC and OA cells, at three different passages and in triplicates, were plated in 96 well white flat bottom plates (COSTAR, Corning, NY, USA) at a density of 10,000 cells/well. The experiment was repeated 3 times in triplicates. Cells were treated with NMRT (5 h in three days); a duplicate control plate was kept at room temperature for the same time. For HDAC inhibitor experiments, trichostatin was added for 1 h prior the application of the HDAC-Glo I/II Assay (Promega, Madison, WI, USA) and in parallel to the fifth and last 1 h NMRT treatment. Performing the assay as described by the manufacturer´s manual, 15–45 min after adding the HDAC-Glo I/II reagent to the cells, luminescence was measured at signal steady-state with the Lumistar microplate luminometer (BMC Labtech, Ortenberg, Germany).

### 4.9. NAD/NADH Detection

HC and OA cells, in triplicates, were plated in 96 well white flat bottom plates (COSTAR) at a density of 10,000 cells/well. For NAD^(+)^/NADH measurements experiment was performed three times at three different passages per cell type. At the end of the 5 h NMRT treatment period over 3 days, the NAD^(+)^/NADH-Glo Detection Reagent of the NAD^(+)^/NADH-Glo Assay and Screening System (Promega) was added to the cells. After a 30 min incubation luminescence was measured with the Lumistar microplate luminometer (BMC Labtech, Ortenberg, Germany). Additionally, NAD^(+)^ or NADH was measured separately (*n* = 3). Sample preparation of the individual measuring set-up was done exactly as described by the manufacturing company in triplicates (Promega). When necessary, IL-1β was added at the beginning of the NMRT treatment in a final concentration of 10 ng/mL.

### 4.10. Annexin/PI Assay

T/C-28a2 cells that have been grown under normoxic and hypoxic conditions were treated with NMRT and tested for changes in healthy, apoptotic and necrotic cell fractions by the use of the Dead Cell Apoptosis Kit (Thermo Fisher Scientific). The detection was performed according to the manufacturers manual and the probes were measured using a flow cytometer (FACSCalibur, BD Bioscience, Franklin Lakes, NJ, USA); data analysis and data acquisition was performed using Cellquest prosoftware (BD Bioscience, USA).

### 4.11. Statistical Analysis

Statistical significance was determined by the two-sample Student’s *t-*test (parametric data) or the Mann–Whitney Rank Sum Test; for multiple group comparison ANOVA with post hoc test was used. *p*-values < 0.05 are considered to be significant (*** *p* < 0.001; ** *p* < 0.01; * *p* < 0.05), all *p*-values are two-sided. Data analysis and display were performed with SigmaPlot 14.0 (Systat Software Inc., San Jose, CA, USA). The number of experiments (*n*) is cited individually in each figure legend; if not pointed out, the vertical error bars drawn in the diagrams indicate the range of fluctuations from which the standard errors were calculated.

## 5. Conclusions

We conclude that NMR modulates the metabolism of chondrocytes via a mild impact. The treatment slightly counteracts OA induced changes in miR expression followed by a minor modulation of downstream regulatory mechanisms (Figure 6). NMRT reduces IL-1β induced catabolic mechanisms and functions as a modulator of the HDACI/II activity. The characterized NAD^(+)^/NADH reduction suggests a possible impact of NMRT on the NAMPT/NAD^(+)^/SIRT axis for chondrocyte protection by restoring normal NADH levels in OA cells. These results propose a beneficial effect for chondrocytes when treated with NMRT which might be even greater under physiologic relevant hypoxic conditions.

## Figures and Tables

**Figure 1 ijms-22-05959-f001:**
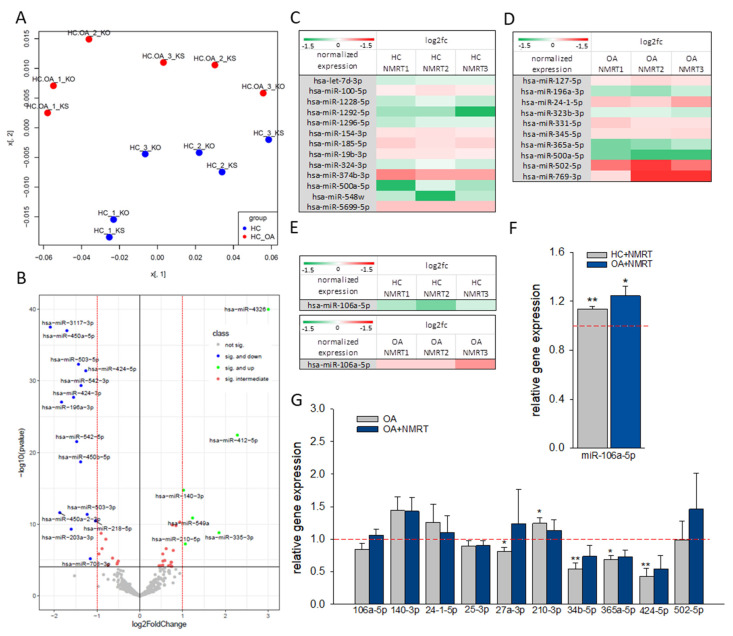
(**A**) Principal component analysis (PCA) based on all detected miRs of all tested samples distinguishes between healthy (HC) and OA chondrocytes; KO: untreated, KS: NMRT 5 h (*n* = 3); x-axis: x[, 1] principle component 1; y-axis: x[, 2] principle component 2. (**B**) Volcano plot representing log2 fold change as a function of the adjusted *p*-value for miR expression in HC versus OA chondrocytes independent of the treatment procedure. The upregulated and downregulated miRs with the highest fold change are particularly designated. The log 1.5-fold change for direct NMRT effects on HC (**C**) or OA (**D**), and especially for miR-106a (**E**) is presented in heat maps with significance of changes at least at the *p* < 0.05 level (Student’s *t*-test). The direct effect of NMRT on HC (grey bars) and OA cells (blue bar) concerning miR-106 expression is depicted (**F**) (untreated cells as calibrator). The bar chart represents the relative gene expression of miRNAs affected under OA and involved in NF-kB signaling by attempting the comparison of HC and OA cells (grey bars); the blue bars present NMRT induced reversed effects when HC cells were compared with OA cells (blue bars) (**G**); in both cases HC cells functioned as calibrator. Data are mean ± SEM (*n* = 5) are given; the Student’s *t*-test functioned as the statistical analysis. *: *p* < 0.05 and **: *p* < 0.01.

**Figure 2 ijms-22-05959-f002:**
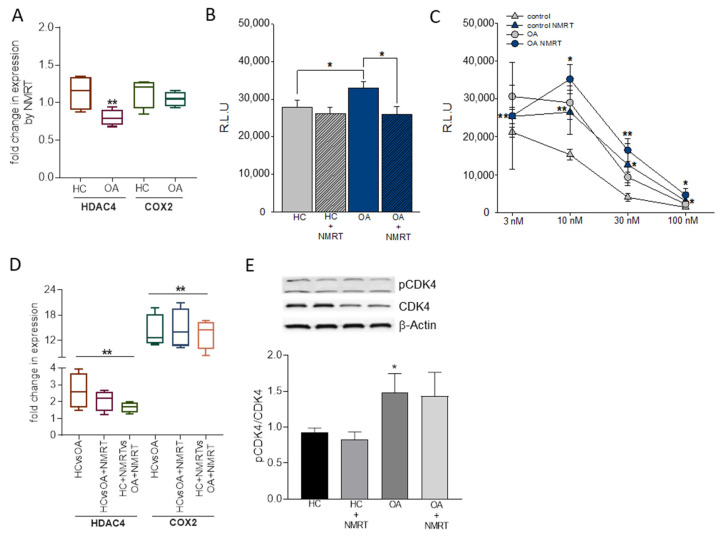
Influence of NMRT on HDAC, COX2 and CDK4 phosphorylation. NMRT induced changes in the expression of HDAC4 and COX2 (**A**,**D**) and the activity (R.L.U., relative luminescence units) of HDAC I/II (**B**) in control (HC) and OA cells are shown. (**A**) summarizes the NMRT induced changes in expression of HC and OA cells (untreated cells functioned as calibrators). Concentration response curve for the inhibition of HDACI/II activity by trichostatin (nM) with or without NMRT treatment in (**C**) (*n* = 3). The box-plot in D represents the difference in expression of HDAC4/COX2 of OA cells (OA) or OA cells treated with NMRT (OA^NMRT^) compared to HC (HC untreated functioned as calibrator; HC/OA; HC/OA^NMRT^); treated OA cells were further compared with NMRT treated HC cells leading to HC^NMRT^/OA^NMRT^ (HC^NMRT^ cells as calibrator); *n* = 4 and each experiment was executed in duplicates. Significant changes compared to the respective calibrator are given: *: *p* < 0.021 and **: *p* = 0.004. CDK4 phosphorylation by western blotting and phospho-CDK4 normalization to CDK4 is shown (**E**). The statistical significance for the bar charts and inhibitor study was evaluated by using the Student’s *t*-test while the differences in median values represented by the box-plots were calculated via the Mann–Whitney Rank Sum test; *: *p* < 0,05; **: *p* < 0,01.

**Figure 3 ijms-22-05959-f003:**
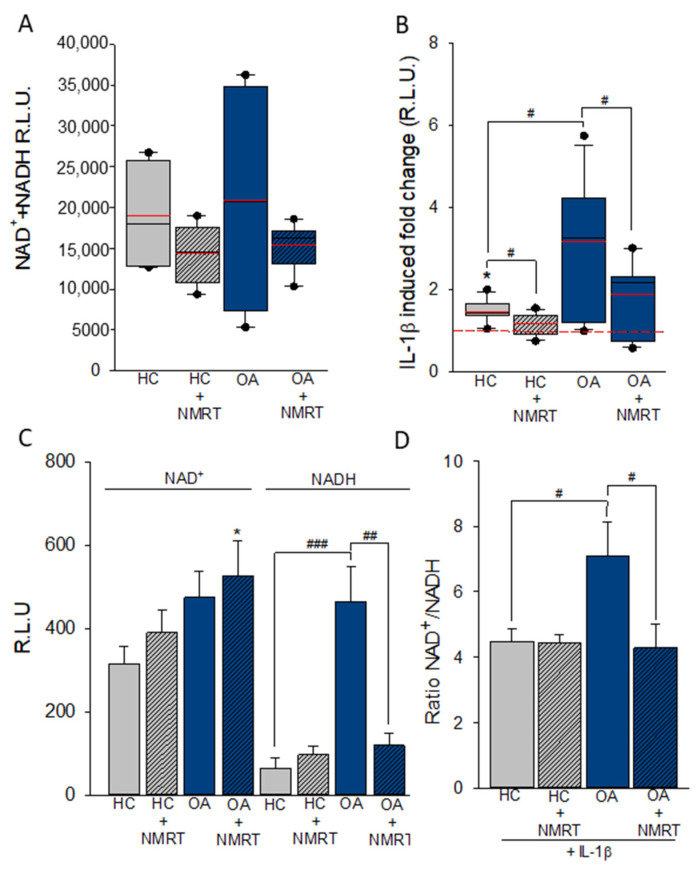
NAD^(+)^ and NADH induced luminescence in HC and OA cells with and without NMR treatment. Box plots with the 5th/95th percentile for HC and OA cells ± NMRT (*n* = 6) (**A**) and the IL-1β induced change under these conditions (**B**) are given. The significant effect by IL-1b in HC cells is indicated by *; # depicts the significant difference between IL-1β effect in HC and OA as well as OA and OA NMRT treated cells. The statistical significances for (**A**,**B**) were evaluated by using the Mann–Whitney Rank Sum Test; the red lines comply with the mean values. NAD^(+)^ and NADH individual measurements are shown (**C**) as well as the NAD^(+)^/NADH ratio in the presence of IL-1β and NMR treatment is being demonstrated (**D**) (*n* = 3). * is the level of significance for NMRT on HC cells, #: significant difference between IL-1β effect in HC and OA as well as OA and OA NMRT treated measured by the two tailed Student’s *t*-test; *,#: *p*<0.05; ##: *p*<0.01; ###: *p*<0.001.

**Figure 4 ijms-22-05959-f004:**
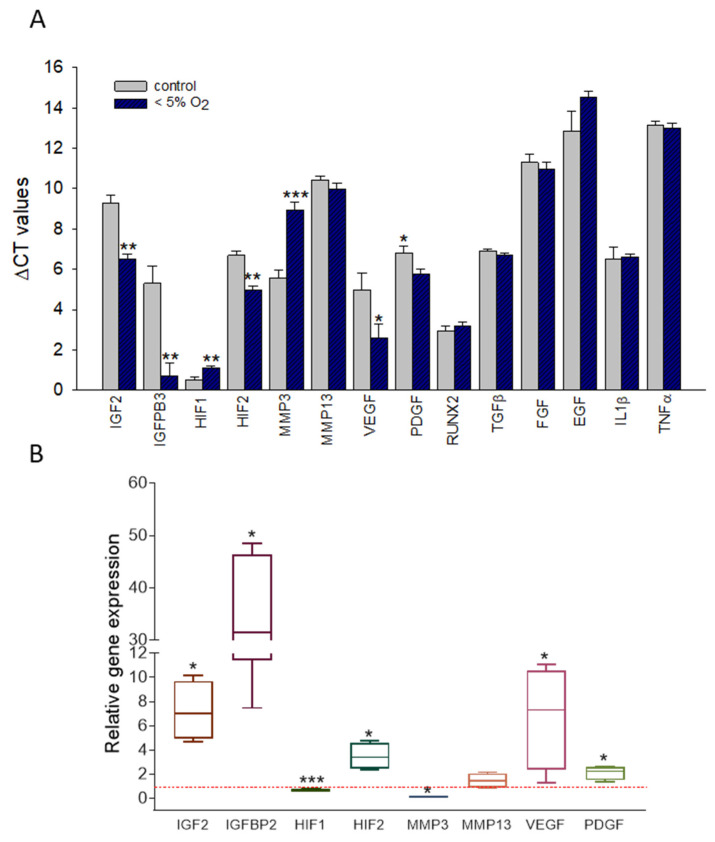
Evaluation of hypoxia induced changes in expression within the chondrocyte test system. The bar chart represents the averaged ∆CT values of specific targets under normoxic and hypoxic conditions (**A**). Statistically significant differences, evaluated by the Student’s *t*-test, are labeled (*: *p* < 0.05; **: *p* < 0.01; ***: *p* < 0.001). Detailed picture of the change in expression induced by O_2_ depletion as ratio (2^−∆∆Ct^) is given within a boxplot–calibrators were cells under normoxic conditions (**B**). Statistical difference was evaluated by the Mann–Whitney Rank Sum Test; *: *p* = 0.029, ***: *p* < 0.001. *n* = 4, measures is duplicates.

**Figure 5 ijms-22-05959-f005:**
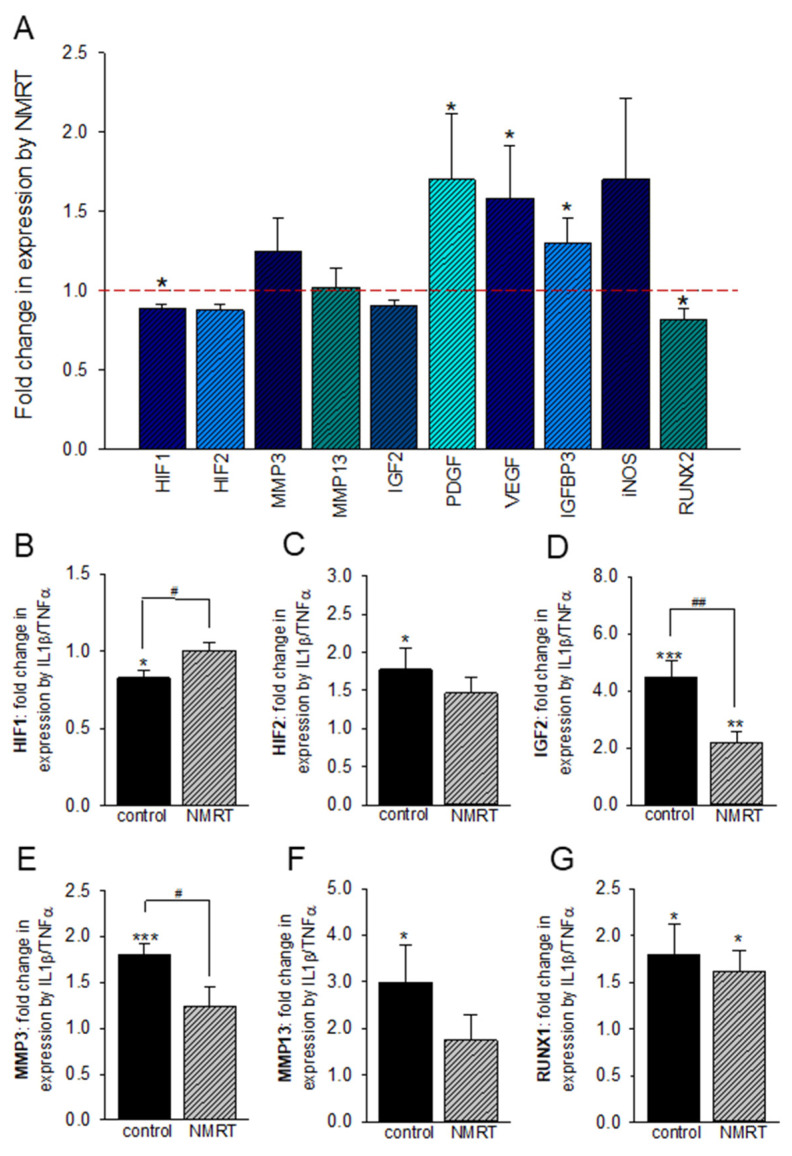
NMRT modulation in the expression of relevant genes in T/C-28a2 cells, the difference between untreated and treated cells is indicated; level of significance (*. *p* < 0.05, *n* = 4) Student’s *t*-test (**A**). Effects of IL-1β/TNFα on the expression of HIF1, HIF2, IGF2, MMP3, MMP13, and RUNX1 are given as change in expression ratio (2^−∆∆Ct^) (**B**–**G**); significant changes are depicted (*) and were calculated via the Student’s *t*-test (*: *p* < 0.05; **: *p* < 0.01 and ***: *p* < 0.001). For HIF1, IGF2, and MMP3 the significance of the IL-1β/TNFα ratio between control and NMRT treatment is outlined by hash tags (#: *p* < 0.05, ##: *p* < 0.01).

**Figure 6 ijms-22-05959-f006:**
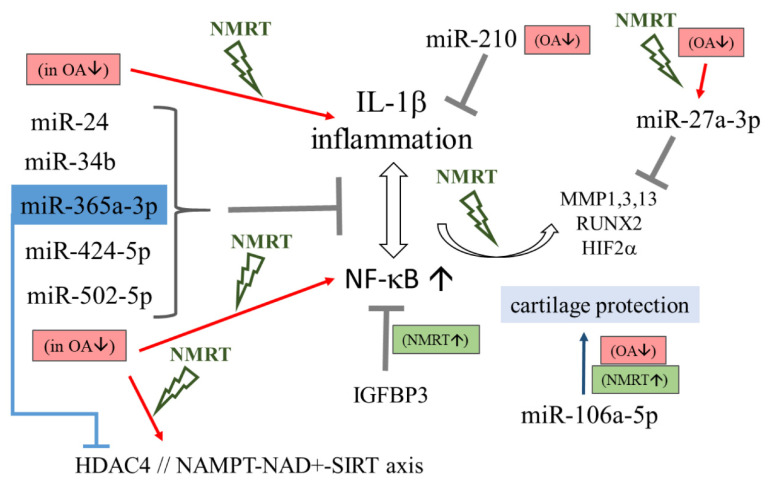
Schematic description of miRs influencing inflammation and NF-ĸB signaling and the gateways where NMRT might interfere. MiRs are listed and their capacity in controlling gene expression under normal conditions as described in the literature and can be deduced from our results, is depicted by the grey T bar. Repression of miRs in OA chondrocytes (OA↓) can intensify inflammation and NF-ĸB activity (red arrows) while these effects were counteracted by NMRT (signed in green); improvements by NMRT are also presented (NMRT↑). The connection between miR-365 and HDAC4 is marked in blue.

## Data Availability

Data is contained within the article or Appendix A.

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
