# Peer review of "Nuclear Magnetic Resonance Therapy Modulates the miRNA Profile in Human Primary OA Chondrocytes and Antagonizes Inflammation in Tc28/2a Cells"

_ijms, 2021, doi:10.3390/ijms22115959_

Round 1

Reviewer 1 Report

In the present manuscript, the authors describe potential beneficial effects of NMRT on human osteoarthritic (OA) or IL-1ß stimulated chondrocytes, in particular under hypoxic conditions. The study mainly focuses on the expression of regulatory microRNAs (miR) and their potential effects, for example on the gene expression of catabolic enzymes and transcription factors, as well as the modulation of HDAC.

This is a very interesting topic. However, the text of the manuscript appears “unfinished” to some extent and requires major revision. Moreover, the results were overestimated and the presentation of the data requires further clarification. Unfortunately, the manuscript is quite confusing and it is hard to enjoy or understand the actually nice study.

The following comments and questions need to be considered/ answered to improve the understanding as well as the overall quality of the manuscript.

1.) The title is not reflecting the study design. Not only primary OA cells, but also T/C-28a2 cells were stimulated with IL-1ß. That the antagonizing effects of NMRT in IL-1b stimualed chondrocytes result from miR profile modulation, is only hypothetical.

2.) In the introduction, some abbreviations were explained (example: matrix metalloproteinases (MMP)), but some were not (COX2, RUNX2, HDAC4). HIF-2a is mentioned in line 80, but the abbreviation is explained in line 89. These are only minor details, but there are also some unclear expressions (line 69: defective programming of cells; possible means kind of a dysfunctional behavior) and further small mistakes (line 37: either “is” associated or usage of commas is required; line 79: comma after “sirtuins” missing), which are distracting. The sentence starting in line 89 has no context to the sentences before (and so on). Overall, the introduction is not well prepared and would benefit from a detailed proofreading.

3.) The results are overestimated to some extent. In line 112, the NMRT effects on the expression of certain miR were described as significant (figure 1 C+D). Are these effects statistically significant or just “strong”? There are no statistically significant changes between the OA and OA+NMRT group in the gene expression analysis (figure 1 G) only between OA and HC (indicated as “*”). Moreover, a 0.2-fold change should not be considered as “significant”. Most effects are just trends. Might a “slight” reduction/ induction of the miR level really cause a significant effect?

4.) Some graphs are confusing and require further explanation or need to be modified.

Figure 2A+D: How was the gene expression calculated? What was the calibrator? This does not look like the ratio (DDCT) as described in the methods part. What means +/- NMRT (were the OA/ HC cells treated or not)? The entire X-axis is unfortunately unclear. How many donors were tested? There are 8 data points but in most of the experiments a maximum of n= 5 was used. Are these technical replicates? These should not be mixed with biological replicates (different donors of primary cells).

5.) The heading 2.4 is not matching well. The topic of the paragraph is not viability (data only provided as supplemental).

6.) Why did the authors use the cell line in the experiments presented in figures 4 and 5? And why were the cells treated with IL-1ß plus TNF, while the cells in the paragraph before only received IL-1ß stimulation? This appears inconsistent.

7.) Figure 4 is confusing. What exactly is demonstrated in B? According to the text (line 202 ff.), these are “the fold changes in expression due to hypoxic conditions …”. According to the figure legend, this is “the effect of NMRT on the expression …”. Which one is correct? Does A (DCT) represent the same as B (DDCT); only a different calculation method? According to the legend, n=4 donors (technical replicates) were used (performed), but there are 4 to 8 data points in graph B.

7.) Figure 5: According to the legend (line 224) DCT values are shown. Is this correct? The effects of IL-1ß + TNF are very low. Usually, the gene expression of MMPs is way higher in response to pro-inflammatory cytokines. But maybe T/C-28a2 cells are less responsive. For how long were the cells stimulated? 24h or longer? (this information is missing in the methods part – or has been overlooked by the reviewer)

What does “significant changes in the ratios” (§) mean? What is the difference to “*”?

8.) The interpretation of the results in the discussion part is overestimated. Of course, statistical significance is not the most important output in science and might also be overestimated, but as mentioned above, some effects on the miR expression are just too low (lower than 0.5-fold). Same is true for the gene expression of MMP13, RUNX2 and HIF2a (line 274), which was not significantly regulated by NMRT. However, there might be a certain trend in MMP13 – but is the effect in HIF2a and RUNX2 really notable?

9.) Some further mistakes: The Y-axis of some graphs have to be corrected (Figure 1 F+G: points, no commas). Line 113: is miR-365a-5p correct or is it miR-365a-3p (as in line 118)?

10.) Further information about the statistics (Which groups were compared in the statistics?) should be included in the figure legends. Moreover, it should be clarified which group served as calibrator to calculate the DDCT (HC, unstimulated cells?).

11.) Are there any information about the sex and age of the OA and healthy donors?  

12.) Suppression of HDAC I/II activity by NMRT (as demonstrated in figure 2A and B) was considered as a beneficial effect (correct?). Might the authors please explain why the opposite effects of NMRT under trichostatin stimulation might induce chondrocyte protection (line 290 ff). Unfortunately, this remains unclear but might be of importance.

Author Response

We thank the reviewers for their helpful, constructive criticism and have revised our manuscript in compliance with their comments.

Our responses to the reviewer’s comments are in italics.

REVIEWER 1:

1.) The title is not reflecting the study design. Not only primary OA cells, but also T/C-28a2 cells were stimulated with IL-1ß. That the antagonizing effects of NMRT in IL-1β stimulated chondrocytes result from miR profile modulation, is only hypothetical.

We have changed the title as follows:

 “Nuclear magnetic resonance therapy modulates the miRNA profile in human primary OA chondrocytes and antagonizes inflammation in Tc28/2a cells”

We hope that this reflects the content of our study as well as possible.

2.) In the introduction, some abbreviations were explained (example: matrix metalloproteinases (MMP)), but some were not (COX2, RUNX2, HDAC4). HIF-2α is mentioned in line 80, but the abbreviation is explained in line 89.

All missing explanations of abbreviations have been added to the introduction.

These are only minor details, but there are also some unclear expressions (line 69: defective programming of cells; possible means kind of a dysfunctional behavior).

Thank you for the hint. The term has been exchanged.

and further small mistakes (line 37: either “is” associated or usage of commas is required; line 79: comma after “sirtuins” missing), which are distracting.

The mistake was corrected by inserting a comma.

The sentence starting in line 89 has no context to the sentences before (and so on). Overall, the introduction is not well prepared and would benefit from a detailed proofreading.

The introduction has been carefully revised. We hope that the new version is more reader-friendly.

3.) The results are overestimated to some extent. In line 112, the NMRT effects on the expression of certain miR were described as significant (figure 1 C+D). Are these effects statistically significant or just “strong”?

Line 104: ….added “statistical significantly changed”…. based on evaluating the significances by using the Student’s t-test showing p values of p<0.05 to p<0.01. A note is added to the figure legend.

There are no statistically significant changes between the OA and OA+NMRT group in the gene expression analysis (figure 1 G) only between OA and HC (indicated as “*”). Moreover, a 0.2-fold change should not be considered as “significant”. Most effects are just trends.

Line 106-111: Thank you for this advice. We changed the manuscript and described the effects outlined in figure 1G as slight and moderate effects.

Might a “slight” reduction/ induction of the miR level really cause a significant effect?

We fully agree with your reasonable doubts. Of course it is hard to believe that an effective modulation by just slight changes in miRNA expression might exist. But based on the view that miRNAs control the fine-tuning within the regulation of gene expression, we deduce that slight effects only requires slight changes in gene expression of specific miRNAs. From our former studies we know that NMRT has a mild impact on the metabolism of chondrocytes, leading to our postulation that tenuous NMRT effects might being controlled by mild miRNA changes.

4.) Some graphs are confusing and require further explanation or need to be modified. Figure 2A+D: How was the gene expression calculated? What was the calibrator? This does not look like the ratio (DDCT) as described in the methods part. What means +/- NMRT (were the OA/ HC cells treated or not)? The entire X-axis is unfortunately unclear. How many donors were tested? There are 8 data points but in most of the experiments a maximum of n= 5 was used. Are these technical replicates? These should not be mixed with biological replicates (different donors of primary cells).

Thank you for this very constructive feedback. In an effort to better understand the presented data we changed part A and D in figure 2. Accordingly, we adapted the figure legend and placed the necessary changes within the result section.

Detail: we changed the comparison of the samples to enhance clarity and after recalculating the data we decided to graph the median values in a box-plot style (incl. max/min whiskers). In A the ratio (2-DDCt) of the NMRT induced change in control (HC) and OA cells are given - this implements the comparison of untreated cells versus treated (NMRT) cells – untreated cells function as calibrator.  Under D by calculating the ratio between control (HC) and OA cells (HC as calibrator) the difference between these two cell types has been demonstrated.  NMRT tendentially reduces the elevated HDAC4 expression in OA cells but did not change the difference in COX2. The difference in HDAC expression was calculated via the ratio between OA cells treated with NMRT and control (HC) cells (HC as calibrator). Finally, if both cell types were treated with NMRT the difference in HDAC expression could be further minimized.

We also apologize for the misleading concerning the number of experiments. We performed four different experiments; each experiment was performed with two different primer pairs per target and was executed in duplicates.

5.) The heading 2.4 is not matching well. The topic of the paragraph is not viability (data only provided as supplemental).

Thank you for this remark. We changed the heading to: “Hypoxia induced changes in the expression of T/C-28a2 cells”

6.) Why did the authors use the cell line in the experiments presented in figures 4 and 5? And why were the cells treated with IL-1ß plus TNF, while the cells in the paragraph before only received IL-1ß stimulation? This appears inconsistent.

The T/C-28a2 cells as a well-established model system to investigate the function of chondrocytes turned out in our hands to be optimally suited for the implementation of the hypoxia model. Due to the fact that the read out, to proof NMRT effectivity, is the modulation of inflammatory effects, using IL-1b and TNF-a to create an inflammatory state was thought to be more effective (Hyc et al., 2003).

Hyc A, Osiecka-Iwan A, Strelczyk P, Moskalewski S: Effect of IL-1β, TNF-agr; and IL-4 on complement regulatory protein mRNA expression in human articular chondrocytes. International Journal of Molecular Medicine (2003); 11(1):91-94.

Due to the fact that our primary chondrocytes derived from end stage OA patients we decided that IL-1b should be sufficient to strengthen the inflammatory status of these types of chondrocytes; based on the fact that primary cells are more difficile in growth we skipped the TNF-a not to put too much stress to the cells.

7.) Figure 4 is confusing. What exactly is demonstrated in B? According to the text (line 202 ff.), these are “the fold changes in expression due to hypoxic conditions …”. According to the figure legend, this is “the effect of NMRT on the expression …”. Which one is correct? Does A (DCT) represent the same as B (DDCT); only a different calculation method? According to the legend, n=4 donors (technical replicates) were used (performed), but there are 4 to 8 data points in graph B.

Thank you for this complementary question. Unfortunately, I must agree that I made a mistake within the figure legend – the fold changes in expression due to hypoxic conditions are shown. In order to gain a better understanding we additionally presented the ΔΔCt values of those targets significantly change in ΔCt value under hypoxic conditions; and yes, it is true that in B data are depicted with a different calculation method to improve the clarity of the presentation.

The figure legend has been corrected and the figure itself has been adapted by replacement of fig 4B by a boxplot diagram.

7.) Figure 5: According to the legend (line 224) DCT values are shown. Is this correct?

We would like to apologize for these unclear indications and descriptions. DCT values are NOT shown – we corrected this mistake and changed the figure legend.

The effects of IL-1ß + TNF are very low. Usually, the gene expression of MMPs is way higher in response to pro-inflammatory cytokines. But maybe T/C-28a2 cells are less responsive.

Thank you for this question! The problem or fact is that IL-1β/TNF-α induce a minor response in respect to MMP expression due to a mandatory high expression of MMPs as an intrinsic characteristics of T/C-28a2 unstimulated cells.

For how long were the cells stimulated? 24h or longer? (this information is missing in the methods part – or has been overlooked by the reviewer)

The cells were stimulated for 3 days – it is mentioned under Material & Methods.

What does “significant changes in the ratios” (§) mean? What is the difference to “*”?

Again my apology for this confusing labelling and our inaccurate data presentation – for clarification we removed the label §.

Figure 5 B-G: The bars represent the averaged ratios deduced from the comparison of untreated and IL-1β/TNF-α treated cells and represent the change in expression by inflammation under control (black) and NMRT (striped); significances are tested by the Student’s t-test with *: p<0.05; **: p<0.01 and ***: p<0.001. Graph has been replaced.

8.) The interpretation of the results in the discussion part is overestimated. Of course, statistical significance is not the most important output in science and might also be overestimated, but as mentioned above, some effects on the miR expression are just too low (lower than 0.5-fold). Same is true for the gene expression of MMP13, RUNX2 and HIF2a (line 274), which was not significantly regulated by NMRT. However, there might be a certain trend in MMP13 – but is the effect in HIF2a and RUNX2 really notable?

We appreciate your concerns and agree that the effects are only mild. According to your recommendation we noted at the beginning of the discussion section that NMRT effects miR expression only mildly.

Based on the diversity and manifoldness of the literature concerning miR and miR-regulation we decided to not specifically change the discussion; only small changes were incorporated. By searching the literature we found the most diverse explanatory approaches of miR-regulation and implemented some of the ideas in the manuscript trying to build a more complex picture - although the NMRT effects were not outrageous.   

9.) Some further mistakes: The Y-axis of some graphs have to be corrected (Figure 1 F+G: points, no commas). Line 113: is miR-365a-5p correct or is it miR-365a-3p (as in line 118)?

The mistakes have been corrected.

10.) Further information about the statistics (Which groups were compared in the statistics?) should be included in the figure legends. Moreover, it should be clarified which group served as calibrator to calculate the DDCT (HC, unstimulated cells?).

We apologize the missing of these informations. By adding the corrections and adjustments to the figures we also described in a more definite way which group served as calibrator.

11.) Are there any information about the sex and age of the OA and healthy donors?

We are sorry not to be able to contribute these informations.

12.) Suppression of HDAC I/II activity by NMRT (as demonstrated in figure 2A and B) was considered as a beneficial effect (correct?). Might the authors please explain why the opposite effects of NMRT under trichostatin stimulation might induce chondrocyte protection (line 290 ff). Unfortunately, this remains unclear but might be of importance.

From the literature we know that HDACs are involved in cartilage and chondrocyte development but also play a crucial role in OA and those HDAC inhibitors (HDACi) can protect cartilage from disease - both therefore representing a potential therapeutic approach against OA. Investigations on the effects of TSA on cartilage degradation in an experimental model of osteoarthritis (OA) revealed that TSA could be considered as a potential agent for treatment for OA. One might conclude quite globally, that inhibiting HDAC activity and increasing TSA function might reduce OA. But on the other hand TSA for instance inhibit cell growth, lead to cell morphology changes, and effectively induce cell apoptosis in a dose-dependent manner in a neural stem cell line.

Obviously the sensitivity against the block by TSA decreases under NMRT. This might be explained by the knowledge that TSA amongst others inhibits HDACs by controlling MAPK activation and NFkB. We know that NMRT influences NFkB and MAPKinases and therefore probably TSA cannot/do not evolve its full inhibitory capacity. In addition, NMRT exerts HDAC reduction upon treatment of cells – subsequent TSA incubation might therefore not be able to demonstrate the same efficacy. 

We hope that we have interpreted the comments correctly and that the modifications of the manuscript will meet the reviewers’ expectations. I remain with best wishes on behalf of all authors,

          Dr. Lohberger Birgit

Reviewer 2 Report

The manuscript analyzes the effect of nuclear magnetic resonance therapy as a modulator for IL-1 induced effects in human osteoarthritic chondrocytes by analysis of miRNA profiles. Effects are compared to human chondrocytes as controls.  

The data set appears complete and justifies the conclusions drawn.

Author Response

The manuscript analyzes the effect of nuclear magnetic resonance therapy as a modulator for IL-1 induced effects in human osteoarthritic chondrocytes by analysis of miRNA profiles. Effects are compared to human chondrocytes as controls.  

The data set appears complete and justifies the conclusions drawn.

We would like to thank you very much for the positive evaluation and are pleased that our work has attracted your interest.

Round 2

Reviewer 1 Report

The present manuscript has benefited significantly from the review process and has been clearly improved! The figures and the text are way more scientific and understandable. Most questions were answered and most comments addressed satisfactorily.

However, there are still some unclear aspects, which have to be addressed. The authors should not hurry and take more time for proper proofreading/ correction.

1.) In figure 2D, the OA group (relative to HC) is compared to the OA+NMRT group (relative to HC+NMRT). This is very complicated and as far as I know, you can’t compare groups based on different calibrators with each other. Has this been discussed with an expert in biostatistics? It really should.

Overall, it remains unclear, why the same data are shown in different graphs (figure 2A and D), using different calculation methods (relative to different calibrators). This is very uncommon and it demonstrates that the NMRT effect just depends on the form of presentation. The comparison between group OA and OA+NMRT (both relative to HC) in figure 2D is statistically correct. Here it is shown, that there is no significant effect due to NMRT stimulation for HDAC4, while there is an effect in figure 2A.

Same is true for COX2; however, the comparison between OA group (relative to HC) and the OA+NMRT group (relative to HC+NMRT) is marked as significant, while the text says (line 144) “COX2 overexpression in OA cells that could not be reversed by NMRT to the level of healthy HC cells”.

2.) There are still commas in the y-axis of the graphs in figure 5.

3.) The authors responded that they analyzed 4 donors (n= 4) but used 2 different primer pairs, which resulted in 8 data points (values). All values were included in the statistics. However, this is statistically incorrect. The values should not be combined and considered as single data points. Maybe a mean value of both primer pairs might be chosen, but these are neither 8 technical nor 8 biological replicates, but rather a mixture of both.

4.) The results and consequently also the conclusions are still overestimated. Line 441: “We   conclude   that   NMR   treatment   counteracts   OA   induced   changes   in   miR expression …” Considering the slight changes and only mild impact on chondrocyte metabolism, the statement is too strong.

Author Response

1.) In figure 2D, the OA group (relative to HC) is compared to the OA+NMRT group (relative to HC+NMRT). This is very complicated and as far as I know, you can’t compare groups based on different calibrators with each other. Has this been discussed with an expert in biostatistics? It really should.

Thank you for your advice and I am sorry that I did not explain it clearer. HCvsOA (HC/OA) marks the comparison between HC cells and OA cells; within the DDCt calculation HC was considered as calibrator and therefore as 1. This comparison should reflect the differences between healthy and OA cells.

Our working hypothesis was that NMRT might provoke OA cells to switch from an OA to a healthier phenotype comparable with HC cells. Therefore, we in addition compared the DCt values of HC cells with the DCt values of OA-cells treated with NMRT by the DDCt method and ratio calculation (HC/OANMRT).

We fully agree that figure 2D might be confusing; to improve the informative value we changed the labeling of the x-axis; the figure legend has been adapted.

Overall, it remains unclear, why the same data are shown in different graphs (figure 2A and D), using different calculation methods (relative to different calibrators). This is very uncommon and it demonstrates that the NMRT effect just depends on the form of presentation.

I apologize for not describing the presented data correctly. In figure 2A HC cells treated with NMRT were normalized to HC (now labeled HCNMRT) and OA cells treated with NMRT were normalized to OA cells (now labeled OANMRT). In figure 2D the comparison between HC cells and OA cells treated or untreated are given.

Figure 2A reflects the effect of NMR while 2D describes the difference between the two cell types plus/minus NMRT with the purpose to describe an effect on chondrocytes in the OA state.

The comparison between group OA and OA+NMRT (both relative to HC) in figure 2D is statistically correct. Here it is shown, that there is no significant effect due to NMRT stimulation for HDAC4, while there is an effect in figure 2A.

The effect in figure 2A describes the direct influence of NMRT on HC and OA cells; HC vs HC+NMRT and OA vs OA+NMRT. We fully agree with the reviewer’s concern pointing to the discrepancy between figure 2A and figure 2D. Although the difference in the HDAC4 expression between HC and OA decreases when OA cells are treated with NMRT, this effect has no statistical significance. This lack in significance might derive from the minor change in OA versus OA+NMRT, although a statistical significance at this point is given.

In addition, it can be considered that even when changes do not highlight statistically the tendency to reduce OA based increase in HDAC4 expression by NMRT is given – especially reflected through the comparison of HC vs OA and HC+NMRT vs OA+NMRT.

We changed the legends of figure 2A/2B to give a clearer picture and adopted the figure legend.

Same is true for COX2; however, the comparison between OA group (relative to HC) and the OA+NMRT group (relative to HC+NMRT) is marked as significant, while the text says (line 144) “COX2 overexpression in OA cells that could not be reversed by NMRT to the level of healthy HC cells”.

For COX2 in figure 2A, there is no influence of NMRT on its expression in HC or OA cells.

In figure 2D: although we detected a significant higher COX2 expression in OA cells when compared to HC cells (HC/OA), this effect could not be minimized either when OA cells were treated with NMRT (HC/OANMRT) nor when both HC and OA cells were treated with NMRT (HCNMRT/OANMRT).

2A represents the NMRT effect on the COX2 expression in HC (HCNMRT) and OA (OANMRT) cells; while in 2D the difference between HC and OA (± NMRT) is depicted.

2.) There are still commas in the y-axis of the graphs in figure 5.

Axis labeling has been corrected.

3.) The authors responded that they analyzed 4 donors (n= 4) but used 2 different primer pairs, which resulted in 8 data points (values). All values were included in the statistics. However, this is statistically incorrect. The values should not be combined and considered as single data points. Maybe a mean value of both primer pairs might be chosen, but these are neither 8 technical nor 8 biological replicates, but rather a mixture of both.

The figures have been adapted and only the data of one primer pair per target are now displayed.

4.) The results and consequently also the conclusions are still overestimated. Line 441: “We   conclude   that   NMR   treatment   counteracts   OA   induced   changes   in   miR expression …” Considering the slight changes and only mild impact on chondrocyte metabolism, the statement is too strong.

 Accordingly, the conclusion has been rewritten.

Round 3

Reviewer 1 Report

The authors nicely addressed the comments and questions of the reviewer. However, there are still two things which need to be revised/ explained.

1.) What does the significance (“**”) refer to in figure 2D? Does the significance refer to all the bars underneath or is it a comparison between HC/OA and HCNMRT/ OANMRT? It rather looks like the latter, but there is no difference between the bars at least in case of COX2.

2.) The authors kind of agreed that it is not correct to include the values of two different primer pairs (one sample, two different values) in the statistics. This was revised in figure 2A + D but not in figure 4B (the bars are similar to the version before). In the first submission, there were 8 data points but only n= 4, implying that this is the same situation as in figure 2. This might also be true for figure 4A and figure 5A-G. The authors should revise all figures.

Author Response

The authors nicely addressed the comments and questions of the reviewer. However, there are still two things which need to be revised/ explained.

1.) What does the significance (“**”) refer to in figure 2D? Does the significance refer to all the bars underneath or is it a comparison between HC/OA and HCNMRT/ OANMRT? It rather looks like the latter, but there is no difference between the bars at least in case of COX2.

The significances refer to all the bars underneath; we apologize for the unclear description. I changed the labelling due to a clearer presentation and added the information to the figure legend.

2.) The authors kind of agreed that it is not correct to include the values of two different primer pairs (one sample, two different values) in the statistics. This was revised in figure 2A + D but not in figure 4B (the bars are similar to the version before). In the first submission, there were 8 data points but only n= 4, implying that this is the same situation as in figure 2.

Thank you for this notice. The qPCR reactions in connection with the experiments under hypoxic conditions were performed with one primer pair per target with n=4 and in duplicates. Nevertheless, the data in both figures were repeatedly controlled, worked out and corrected. The statistics for the boxplot in figure 4b was once more gained by using the Mann-Whitney Rank Sum Test – figure legends were adapted. The display in figure 5 is unchanged due to the fact that the given results demonstrate the summary of four experiments performed in duplicates and with just one pair of primer.

Round 4

Reviewer 1 Report

The authors nicely addressed all comments and questions. There are no further comments.